# *Neuropeltis acuminata* (P. Beauv.): Investigation of the Chemical Variability and In Vitro Anti-inflammatory Activity of the Leaf Essential Oil from the Ivorian Species

**DOI:** 10.3390/molecules27123759

**Published:** 2022-06-10

**Authors:** Didjour Albert Kambiré, Ahmont Claude Landry Kablan, Thierry Acafou Yapi, Sophie Vincenti, Jacques Maury, Nicolas Baldovini, Pierre Tomi, Mathieu Paoli, Jean Brice Boti, Félix Tomi

**Affiliations:** 1UPR de Chimie Organique, Département de Mathématiques, Physique et Chimie, UFR des Sciences Biologiques, Université Péléforo Gon Coulibaly, Korhogo BP 1328, Côte d’Ivoire; dakambire@gmail.com (D.A.K.); kablanahmont@yahoo.fr (A.C.L.K.); 2Laboratoire Sciences Pour l’Environnement, Université de Corse—CNRS, UMR 6134 SPE, Route des Sanguinaires, 20000 Ajaccio, France; vincenti_s@univ-corse.fr (S.V.); maury_j@univ-corse.fr (J.M.); tomi_p@univ-corse.fr (P.T.); paoli_m@univ-corse.fr (M.P.); 3Laboratoire de Constitution et Réaction de la Matière, UFR-SSMT, Université Félix Houphouët-Boigny, Abidjan BP V34, Côte d’Ivoire; acafouth@yahoo.fr (T.A.Y.); jeanbriceboti@hotmail.fr (J.B.B.); 4Institut de Chimie de Nice, CNRS UMR 7272, Université Côte d’Azur, Parc Valrose, CEDEX 2, 06108 Nice, France; nicolas.baldovini@unice.fr

**Keywords:** *Neuropeltis acuminata*, leaf essential oil, chemical variability, anti-inflammatory activity, δ-cadinen-11-ol

## Abstract

The variability of chemical composition of the leaf essential oil (EO) from *Neuropeltis acuminata*, a climbing liana growing wild in Ivory Coast, was investigated for the first time. The in vitro anti-inflammatory activity was also evaluated. Thirty oil samples were isolated from leaves collected in three forests of the country and analyzed using a combination of Column Chromatography (CC), Gas Chromatography with Retention Indices (GC(FID)), Gas Chromatography-Mass Spectrometry (GC-MS), and ^13^Carbon-Nuclear Magnetic Resonance (^13^C-NMR). Fractionation by CC led to the first-time isolation from natural source of δ-cadinen-11-ol, whose structural elucidation by one dimension (1D) and 2D-NMR spectroscopy is reported here. Finally, 103 constituents accounting for 95.7 to 99.6% of the samples’ compositions were identified. As significant variations of the major constituents were observed, the 30 oil compositions were submitted to hierarchical cluster and principal components analyses. Five distinct groups were evidenced: Group I, dominated by (*E*)-β-caryophyllene, kessane, and δ-cadinene, while the main constituents of Group II were germacrene B, ledol, α-humulene, (*E*)-γ-bisabolen-12-ol, and γ-elemene. Group III exhibited guaiol, germacrene D, atractylone, (*E*)-γ-bisabolen-12-ol, δ-cadinene and bulnesol as main compounds. Group IV was dominated by (*E*)-nerolidol, guaiol, selina-4(15),7(11)-diene and bulnesol, whereas (*E*)-β-caryophyllene, α-humulene and α-muurolene were the prevalent compounds of Group V. As the harvest took place in the same dry season in the three forests, the observed chemical variability could be related to harvest sites, which includes climatic and pedologic factors, although genetic factors could not be excluded. The leaf oil sample S24 behaved as a high inhibitor of LipOXygenase (LOX) activity (half maximum Inhibitory Concentration, IC_50_: 0.059 ± 0.001 mg mL^−1^), suggesting an anti-inflammatory potential.

## 1. Introduction

The genus *Neuropeltis* of the Convolvulaceae family, comprises about thirteen species, of which nine are restricted to the Guineo-Congolian region of Africa. The remaining four species occur in tropical Asia [1,2]. *Neuropeltis acuminata* (P. Beauv.) is a woody liana with contorted stems up to 40 m long and 25 cm in diameter. Its leaves are alternate, simple with acuminated apex and entire margin. The inflorescences are axillary, and the flowers are bisexual, regular, and fragrant. The fruits of *N. acuminata* are rounded capsules of 7 mm in diameter, surrounded at the base by persistent calyx and enlarged bract. This species is widely distributed from Senegal to the Central African Republic, Democratic Republic of Congo, and Angola and located mainly in evergreen primary forests. In older forests, it becomes one of the dominant climbing species. The stems of *N. acuminata* are used as ropes and for tying in the construction of houses in Ghana and Ivory Coast, while the leaves are eaten as a vegetable in Gabon [1,2,3,4,5]. In South Cameroon, the fibers of the plant were used for a long time as dish sponges and toilet gloves [6].

The micromechanical, physicochemical, and thermal properties of *N. acuminata* fibers were evaluated through some studies reported in the literature. A few functional groups attributable to cellulose, hemicellulose, polysaccharides, lignins, and aromatic compounds were evidenced by infrared spectroscopy applied to fibers extracts [6,7]. However, no in-depth investigation on the identification of individual chemical constituents from solvent extracts and essential oils of *N. acuminata* has so far been reported in the literature.

The present study is part of the continuation of the chemical characterization of aromatic and medicinal plants from Ivory Coast [8,9,10,11,12,13,14,15,16]. It aims to determine for the first time the variability of the chemical composition of the leaf EO of *N. acuminata*. Fractionation by CC led to the first-time isolation from natural source of δ-cadinen-11-ol, whose structural elucidation by 1D and 2D-NMR spectroscopy is reported here. An EO sample was also tested for its in vitro anti-inflammatory activity. 

## 2. Results and Discussion

Thirty samples of fresh leaves of *N. acuminata* were harvested in the Bossématié forest (Eastern Ivory Coast), the Haut-Sassandra forest (Western Ivory Coast) and the Yapo-Abbé forest (Southern Ivory Coast) Appendix A. The EO samples were isolated by hydrodistillation, and the extraction yields calculated on weight basis (*w*/*w*) varied poorly (Yield: 0.58–1.03%). The higher value of yield (Mean: 0.92%) was observed in the Bossématié forest and the lower (Mean: 0.73%) was observed in the Haut-Sassandra forest (Station 4) and the Yapo-Abbé forest (Station 5). Combination of chromatographic and spectroscopic techniques (GC(RI), GC-MS, and ^13^C-NMR) was used to perform the analyses. A computerized ^13^C-NMR method developed at the University of Corsica was applied to the oil samples. This method allows identification of components present at a content as low as 0.4–0.5% and compiled in our laboratory-made ^13^C-NMR spectral data library [17,18]. Various constituents were identified by the three above techniques.

Compounds bearing the germacrane and elemane skeletons needed special attention. Indeed, under thermal GC and GC-MS conditions, germacrene compounds bearing the cyclodeca-1,5-diene sub-structure were partially or totally rearranged to the corresponding elemenes (sigmatropic [3,3] rearrangement) [19]. As germacrene B and γ-elemene on the one hand, then furanodiene and curzerene on the other hand, were detected by GC-MS and ^13^C-NMR, their correct contents were obtained by combination of GC(FID) and ^13^C-NMR [15]. However, δ-elemene, β-elemene, and curzerenone were actual secondary metabolites produced by the plant and not rearranged products since germacrene C, germacrene A, and furanodienone were not detected by ^13^C-NMR. Heat-sensitive compounds are one of the main causes of artifact formation during GC and GC-MS analyses. Hence, the combination of different analytical techniques is highly recommended for a better qualitative and quantitative analysis of essential oils.

### 2.1. Detailed Analysis of Essential Oil Samples S3, S6, S13, S20, and S24

Despite the use of complementary techniques, several compounds remained unidentified. Therefore, samples S3 (2.605 g), S6 (3.280 g), S13 (2.318 g), S20 (2.148 g), and S24 (4.110 g), which displayed different chemical profiles and contained minor unidentified constituents, were separately submitted to repetitive fractionation by silica gel CC. These samples also represented the different stations of the three sampling forests. Analysis of CC fractions by GC(RI), GC-MS, and ^13^C-NMR led to the identification of several minor components. In parallel, five compounds whose NMR data were not compiled in our laboratory spectral database were isolated, and their structures were elucidated by 1D- and 2D-NMR analyses. After structural elucidations, ^13^C-NMR data of four compounds among the five, were found in the literature. These data were very close to our experimental ^13^C-NMR data, confirming the proposed structures: 1,11-oxidocalamenene (**49**) isolated from fraction F4.1.1 (19 mg, 94.5%) of sample S20; atractylone (**93**) from sample S24, fraction F3.1.2.1 (38 mg, 98.7%); (*E*)-γ-bisabolen-12-al (**101**) from sample S24, fraction F3.1.2.2 (27 mg, 96.3%) and (*E*)-γ-bisabolen-12-ol (**102**), fraction F6.3.4.1 (41 mg, 97.4%) of sample S24 (Figure 1, Table 1) [20,21,22,23]. As the ^13^C-NMR and 2D-NMR data of the fifth compound (δ-cadinen-11-ol (**96**)) were not found in the literature, we reported its structural elucidation.

#### 2.1.1. Isolation and Structural Elucidation of δ-Cadinen-11-ol

Repetitive silica gel column chromatography, performed on sample S6, led to the isolation of compound **96** (GC: 96.7%; RIs apol/pol: 1651/2271) in the sub-fraction F.5.4.2 (12 mg). The electron impact (EI) mass spectrum of compound **96** showed a molecular ion peak (M^•+^) at *m/z* = 220 and an M^•+^ − 18 peak (*m/z* = 202), characteristic of a sesquiterpene alcohol. The measured exact mass was 220.1829 g/mol, corresponding to C_15_H_24_O formula (calculated mass = 220.1827 g/mol). The ^1^H-NMR, ^13^C-NMR, and DEPT spectra agreed with this formula, which involved four unsaturation degrees (Table 2) Appendix A. The alcohol function was confirmed by the quaternary oxide carbon (C11, 74.57 ppm). The presence of four sp^2^ carbon signals, involved in two C=C double bonds, demonstrated that compound **96** obviously bears a bicyclic structure. The ^1^H-NMR spectrum evidenced two methyl groups linked to sp^2^ quaternary carbons (H14, 1.64 ppm, broad s; H15, 1.67 ppm, broad s) and an ethylenic proton (H5, 5.71 ppm, hept: 1.5 Hz) belonging to tri- and tetrasubstituted double bonds. Correlations observed in the COSY spectrum evidenced two proton groups formed by the sequences H2-H3 and H5-H6-H7-H8-H9. The HMBC correlations plots were used to build the bicyclic skeleton by linking together the observed proton groups. Indeed, H15 was correlated to C3, C4, and C5, while proton H7 (1.41 ppm, ddd: 11.0, 8.1, 3.1 Hz) correlated to C11, C12 and C13, appeared very informative about the location of the hydroxyl group (2-hydroxy-isopropyl). Correlations of methyl protons H14 with C1, C9, and C10 on the one hand, added to plots between H2α, H2β, and C1, C3, C4, C6, and C10 on the other hand, evidenced a cadinadiene alcohol structure. The relative stereochemistry of carbons C6 and C7 was established through NOESY spatial correlations. Protons H6, H12, and H13 correlated together indicating a *cis* stereochemistry of H6 and the 2-hydroxy-isopropyl group. Finally, the absence of correlations plots between H6 and H7 made it possible to determine the structure of compound **96** as *δ*-cadinen-11-ol (Figure 1). This compound is isolated from natural source for the first time. It was previously mentioned in the literature as a synthetic derivative and only its ^1^H-NMR data were described. The 1D- and 2D-NMR data of this compound are reported in Table 2. 

#### 2.1.2. Chemical Composition of Leaf Essential Oil Samples S3, S6, S13, S20 and S24

The chemical composition of samples S3, S6, S13, S20, and S24, which displayed different chemical profiles, were determined by combination of chromatographic and spectroscopic techniques. A total of 103 compounds, representing respectively 95.7, 97.9, 96.8, 99.6, and 98.1% of the samples’ whole compositions, were identified (Table 3). The five oil samples were very rich in sesquiterpenes (90.2–97.5%). Obviously, the major compounds displayed significant variations. Indeed, sample S3 was dominated by germacrene B (15.5%) and ledol (12.5%), followed by (*E*)-γ-bisabolen-12-ol (6.1%), α-humulene (5.8%), γ-elemene (5.5%) and kessane (5.2%). The main components of sample S6 were (*E*)-nerolidol (23.3%) and guaiol (12.2%), followed by selina-4(15),7(11)-diene (8.3%), β-eudesmol (6.3%) and bulnesol (5.2%). (*E*)-β-caryophyllene (20.0%), kessane (11.5%), δ-cadinene (8.4%), cadina-1(10),4-dien-8α-ol (8.1%), and germacrene D (5.2%), were the predominant constituents of sample S13. Sample S20 was largely dominated by (*E*)-β-caryophyllene (34.4%) and α-humulene (27.1%), followed by α-muurolene (6.6%) and β-elemene (4.6%), whereas guaiol (11.6%), germacrene D (10.2%), atractylone (10.0%), (*E*)-γ-bisabolen-12-ol (7.1%), δ-cadinene (7.0%), and bulnesol (6.2%), were the main compounds of sample S24. The five determined chemical compositions exhibited qualitative and quantitative variations.

### 2.2. Chemical Variability of Leaf Essential Oil from N. acuminata

The evaluation of the chemical variability of leaf EO from *N. acuminata* was conducted on 30 samples collected in three forests during the dry season: Bossématié forest (Eastern Ivory Coast, four samples from one station), Haut-Sassandra forest (Western Ivory Coast, seventeen samples from three stations), and Yapo-Abbé forest (Southern Ivory Coast, nine samples from two stations). Whatever the sample, *N. acuminata* can be considered as sesquiterpene-rich EO plant (87.1–98.1%). However, the main compounds varied drastically from sample to sample: (*E*)-β-caryophyllene (0.9–45.4%), kessane (0.1–32.5%), α-humulene (0.5–31.2%), (*E*)-nerolidol (0.2–30.8%), germacrene B (0.7–16.6%), germacrene D (0.5–14.4%), guaiol (0.3–13.8%), ledol (0.1–13.2%), atractylone (0.4–12.1%), cadina-1(10),4-dien-8α-ol (0.1–12.1%), δ-cadinene (0.4–12.0%), selina-4(15),7(11)-diene (0.1–11.2%), and α-muurolene (0.1–10.9%). Thus, statistical analyses were performed on the 30 EO compositions to emphasize chemical variability.

Five distinct groups were observable on the dendrogram from the hierarchical cluster analysis (HCA): Group I (8 samples), Group II (4 samples), Group III (9 samples), Group IV (5 samples), and Group V (4 samples) (Figure 2). The five groups from HCA were also evidenced by the principal component analysis (PCA) map of samples distribution (Figure 3). Its principal factors F1 and F2 accounted for 68.84% of the total variance of the chemical composition. Table 4 contains the mean contents (M) and the standard deviations (SD) of the main compounds that discriminated the groups on PCA.

Group I was dominated by (*E*)-β-caryophyllene (M = 17.6%, SD = 4.2%) and kessane (M = 10.3%, SD = 9.9%), followed by δ-cadinene (M = 6.2%, SD = 3.4%), germacrene D (M = 4.3%, SD = 1.1%), guaiol (M = 4.2%, SD = 2.5%), cadina-1(10),4-dien-8α-ol (M = 3.6%, SD = 4.8%), ledol (M = 3.4%, SD = 3.1%), and germacrene B (M = 3.4%, SD = 0.8%). Kessane and cadina-1(10),4-dien-8α-ol were, respectively, absent from the samples of Groups III and II. The main constituents of Group II were germacrene B (M = 13.4%, SD = 3.1%), ledol (M = 10.6%, SD = 2.7%), α-humulene (M = 6.6%, SD = 2.1%), (*E*)-γ-bisabolen-12-ol (M = 5.8%, SD = 0.4%), γ-elemene (M = 5.0%, SD = 0.8%), kessane (M = 4.8%, SD = 0.6%), and β-elemene (M = 4.6%, SD = 0.5%). Groups I and II were characterized by high contents of sesquiterpene hydrocarbons (respectively M = 58.6 and 55.8%, vs. 34.5 and 30.7% of oxygenated sesquiterpenes). In contrast, Groups III and IV were composed by oxygenated sesquiterpenes-rich oils (respectively M = 49.1 and 67.7%, vs. 42.1 and 27.0% of sesquiterpene hydrocarbons). Indeed, Group III exhibited guaiol (M = 11.4%, SD = 2.0%), germacrene D (M = 10.6%, SD = 1.6%), atractylone (M = 9.6%, SD = 2.3%), (*E*)-γ-bisabolen-12-ol (M = 7.3%, SD = 1.3%), δ-cadinene (M = 7.3%, SD = 0.4%), bulnesol (M = 5.9%, SD = 1.2%), and furanodiene (M = 4.0%, SD = 1.2%), as major compounds, while Group IV was dominated by (*E*)-nerolidol (M = 23.8%, SD = 5.4%), guaiol (M = 12.4%, SD = 1.3%) and selina-4(15),7(11)-diene (M = 9.5%, SD = 1.7%), followed by bulnesol (M = 6.5%, SD = 1.1%), β-elemol (M = 4.5%, SD = 0.5%), and (*E*)-γ-bisabolen-12-ol (M = 4.4%, SD = 2.6%). The last group (Group V) was widely dominated by sesquiterpene hydrocarbons (M = 75.3%, vs. 20.6% of oxygenated sesquiterpenes). (*E*)-β-caryophyllene (M = 36.6%, SD = 5.9%), α-humulene (M = 18.9%, SD = 12.1%), and α-muurolene (M = 6.2%, SD = 3.5%) were its prevalent compounds. Therefore, it could be stated that the chemical composition of the leaf EO displayed qualitative and quantitative variability.

The 8 samples of Group I were harvested from station 2 of the Haut-Sassandra forest, while the 4 samples of Group II were collected in the Bossematié forest. Group III is constituted by the 9 samples from the Yapo-Abbé forest. Samples from Groups IV and V were respectively harvested at stations 3 and 4 of the Haut-Sassandra forest. As the harvest took place in the same dry season in the three forests, the observed chemical variability could be related to harvest sites, which include climatic and pedologic factors, then vegetative stage (young or old lianas), although genetics factors could not be excluded.

### 2.3. Evaluation of In Vitro Anti-Inflammatory Activity

The in vitro anti-inflammatory activity of *N. acuminata* leaf EO (S24) was evaluated by the LOX inhibition method. Indeed, LOXs are a nonheme iron-containing dioxygenases, which were responsible for the formation of biologically active metabolites. They were key enzymes in the biosynthesis of leukotrienes that were mediators of many disorders related with inflammatory processes such as arthritis, bronchial asthma, and cancer [33,34,35,36]. The discovery of novel LOX inhibitors appeared as crucial point because they would prevent overproduction of leukotrienes and thus could constituted new therapeutic tools for treating of human inflammation-related diseases.

The inhibition ability of soybean LOX by S24 was measured and considered as an indicator of its potential anti-inflammatory activity. Results of LOX inhibition tests were presented in Table 5. The *N. acuminata* leaf essential oil inhibited LOX activity and this inhibition increased with the concentration of the oil (15.20% at 0.0125 mg mL^−1^ up to 81.87% at 0.100 mg mL^−1^). The IC_50_ values were calculated for S24 and for inhibitor NorDihydroGuaiaretic Acid (NDGA), a non-competitive inhibitor of lipoxygenase usually used as reference in anti-inflammatory assays (Table 5) [34,35,36]. The IC_50_ value of S24 (0.059 ± 0.001 mg mL^−1^) was only 4.5-higher than IC_50_ value of NDGA (0.013 ± 0.003 mg mL^−1^). This low ratio between the two IC_50_ values (S24 vs. NDGA) allowed to consider the *N. acuminata* leaf essential oil as a high inhibitor of the LOX activity, suggesting an anti-inflammatory potential [37].

## 3. Materials and Methods

### 3.1. Plant Material

Fresh leaves samples from *N. acuminata* were collected at six stations of three forests: Bossématié forest, Region of Abengourou, Eastern Ivory Coast (Station 1); Haut-Sassandra forest, Western Ivory Coast (Stations 2–4); Yapo-Abbé forest, Southern Ivory Coast (Stations 5 and 6). Geographical coordinates: Station 1 (6°29′26.0″ N and 3°29′11.7″ W), Station 2 (6°53′40.2″ N and 6°55′36.3″ W), Station 3 (6°57′08.5″ N and 6°59′00.5″ W), Station 4 (6°54′52.7″ N and 6°57′21.1″ W), Station 5 (5°41′08.0″ N and 4°06′31.7″ W) and Station 6 (5°41′48.7″ N and 4°05′31.0″ W). The harvest took place during the dry season (January and February 2021). Plant material was authenticated by botanists from Centre Suisse de Recherches Scientifiques (CSRS) and Centre National de Floristique (CNF) Abidjan, Ivory Coast. A voucher specimen was deposited at the herbarium of CNF, Abidjan, with the reference LAA 11029.

### 3.2. Essential Oil Isolation and Fractionation

The EO samples were extracted by hydrodistillation of fresh leaves for 3 h each, using a Clevenger-type apparatus. The extraction yields were calculated using the weight of essential oil/weight of fresh leaves ratio (*w*/*w*) and reported in Appendix A. Five oil samples S3 (2.605 g), S6 (3.280 g), S13 (2.318 g), S20 (2.148 g), and S24 (4.110 g) were chromatographed on column with respectively 95 g, 100 g, 90 g, 80 g, and 130 g of silica gel (Acros Organics, Geel, Belgium, 60–200 μm). For each sample, seven fractions were eluted with a mixture of solvents, *n*-pentane (P) (VWR Chemicals, 99%)/diethyl ether (DE) (VWR Chemicals, 99%) on increasing polarity (P/DE 100/0 to 0/100). F1 and F2 (100% P) contained olefins; F3–F6 contained medium polar compounds while F7 (100% DE) contained polar compounds. The mass of fractions were reported in the Table 6. Repetitive column chromatography (SiO_2_, 35–70 μm, Acros Organics, Geel, Belgium) were performed on fractions from samples S6 (F4 and F5), S20 (F4 and F5), and S24 (F3 and F6), in order to isolate several unidentified compounds.

### 3.3. Gas Chromatography

Analyses were performed on a Clarus 500 PerkinElmer Chromatograph (PerkinElmer, Courtaboeuf, France), equipped with flame ionization detector (FID) and two fused-silica capillary columns (50 m × 0.22 mm, film thickness 0.25 µm), BP-1 (polydimethylsiloxane), and BP-20 (polyethylene glycol). The oven temperature was programmed from 60 °C to 220 °C at 2 °C/min and then held isothermal at 220 °C for 20 min; injector temperature: 250 °C; detector temperature: 250 °C; carrier gas: hydrogen (0.8 mL/min); split: 1/60; injected volume: 0.5 µL. Retention indices (RI) were calculated relative to the retention times of a series of *n*-alkanes (C8–C29) with linear interpolation (“Target Compounds” software from PerkinElmer, Courtaboeuf, France). The quantification of volatile compounds was obtained using Relative Response Factor (RFF), calculated according to the International Organization of the Flavor Industry (IOFI) [32]. The relative proportion of each compound (expressed in g/100 g) was calculated using the amount of EO and reference (Methyl octanoate), peak area and relative response factors.

### 3.4. Gas Chromatography-Mass Spectrometry in Electron Impact Mode

Analyses were performed on Clarus SQ8S PerkinElmer TurboMass detector (quadrupole), directly coupled with a Clarus 580 PerkinElmer Autosystem XL (PerkinElmer, Courtaboeuf, France), equipped with a BP-1 (polydimethylsiloxane) fused-silica capillary column (50 m × 0.22 mm i.d., film thickness 0.25 µm). The oven temperature was programmed from 60 to 220 °C at 2°/min and then held isothermal for 20 min; injector temperature, 250 °C; ion-source temperature, 250 °C; carrier gas, Helium (1 mL/min); split ratio, 1:80; injection volume, 0.5 µL; ionization energy, 70 eV. The electron ionization (EI) mass spectra were acquired over the mass range 35–350 Da.

### 3.5. Gas Chromatography-High Resolution Mass Spectrometry

High-resolution EI-mass spectra were performed on Agilent 7200 GC-QTOF system (Agilent, Santa Clara, CA, USA) equipped with an Agilent J&W, VF-waxMS capillary column (30 m × 0.25 mm; 0.25 µm film thickness). The mass spectrometer was operating at 70 eV with an acquisition rate of 2 GHz over a 35−450 *m/z* range, affording a resolution of ∼8000. Injection volume 1 µL; split ratio 1:20; inlet temperature 250 °C, detector temperature 230 °C; column flow (Helium) 1.2 mL/min; temperature program for oven 60 °C (5 min isotherm) to 240 °C at 5 °C/min, then 10 min isotherm at 240 °C.

### 3.6. Nuclear Magnetic Resonance

All spectra were recorded on a Bruker AVANCE 400 Fourier transform spectrometer (Bruker, Wissembourg, France) operating at 400.132 MHz for ^1^H and 100.623 MHz for ^13^C, equipped with a 5 mm probe. Solvents used were CDCl_3_ and C_6_D_6_, with all shifts referred to internal TMS. The ^1^H-NMR spectra were recorded with the following parameters: pulse width (PW), 4.3 µs; relaxation delay 1 s and acquisition time 2.6 s for 32 K data table with a spectral width (SW) of 6000 Hz. ^13^C-NMR spectra of the oil samples and fractions of CC were recorded with the following parameters: pulse width = 4 µs (flip angle 45°); relaxation delay D1 = 0.1 s, acquisition time = 2.7 s for 128 K data table with a spectral width of 25,000 Hz (250 ppm); CPD mode decoupling; digital resolution = 0.183 Hz/pt. The number of accumulated scans was 3000 for each sample or fraction (40 mg, when available, in 0.5 mL of CDCl_3_ or C_6_D_6_). For the 2D spectra, sequences from Bruker Topspin^TM^ (Bruker, Wissembourg, France) library (DEPT, COSY, HMBC and NOESY) and Gradient-enhanced sequences were used. 1D and 2D Spectra were processed via MestreNOVA software (version 12.0.0-20080, Mestrelab, Santiago de Compostela, Spain).

### 3.7. Identification of Individual Components

Identification of the individual components was based on (i) comparison of their GC retention indices on apolar and polar columns, with those of reference compounds [24,38]; (ii) computer search using digital libraries of mass spectral data [38,39,40]; (iii) ^13^C-NMR spectroscopy following a computerized method developed in our laboratory using a home-made software by comparison of the chemical shift values in EO or fraction spectrum with those of reference spectra compiled in the laboratory-built library [15,17]. In the investigated samples, individual components were identified by ^13^C-NMR at contents as low as 0.4–0.5%. A few compounds were identified by comparison with literature data.

### 3.8. Statistical Analysis

Data of the 30 investigated samples of *N. acuminata* were submitted to hierarchical cluster analysis (HCA) and principal component analysis (PCA) using XLSTAT software (Addinsoft, Paris, France) [41]. Only constituents in a concentration higher than 1.0% were used as variables for the PCA analysis.

### 3.9. In Vitro Anti-Inflammatory Capacity of Neuropeltis acuminata Leaf Essential Oil

The in vitro anti-inflammatory capacity of *N. acuminata* leaf EO (S24) was conducted by in vitro lipoxygenase inhibition assay [42,43,44]. Lipoxidase type I-B (Lipoxygenase, LOX, EC 1.13.11.12) from soybean purchased from Sigma-Aldrich (Saint-Quentin-Fallavier, France) was used for the in vitro analysis for LOX inhibitory activity. It was determined by continuously monitoring the formation of conjugated dienes of the 13-hydroperoxides of linoleic acid at 234 nm using a spectrophotometric method [42,43,44].

The LOX solution was prepared by dissolving around 5.7 units mL^−1^ of LOX in PBS (Phosphate Buffer Solution; 1 unit corresponding to 1 µmol of hydroperoxide formed per min). The S24 sample diluted in dimethyl-sulfoxide (DMSO) was used as inhibitor solution for LOX inhibition activity assay. Five concentrations were tested: 0.0125, 0.0250, 0.0500, 0.0800 and 0.1000 mg/mL.

The LOX inhibition assays were performed as previously described [44]. Briefly, 10 µL of LOX solution and 10 µL of inhibitor solution were mixed in 970 µL of boric acid buffer (50 mM; pH 9.0) and incubating them briefly at room temperature. The enzymatic reaction started by addition of 10 µL of substrate solution (Linoleic acid, 25 mM) and the reaction rate was recorded for 30 s at 234 nm. One measurement was carried out in the absence of inhibitor solution and another with DMSO mixed with distilled water (99.85% DMSO in distilled water) in order to evaluate a possible inhibitory effect of DMSO. LOX activity was not affected by DMSO and the measurement of the LOX activity without inhibitor solution was considered as control (100% enzymatic reaction). All measurements were performed in triplicate. The percentage of LOX inhibition was calculated according to the equation:Inhibition % = (V_control_ − V_S24_) × 100/V_control_V_control_ is the activity of LOX in absence of inhibitor solution and V_S24_ is the activity of LOX in presence of inhibitor solution [45]. The IC_50_ was calculated by the concentration of S24 in DMSO inhibiting 50% of LOX activity.

### 3.10. Spectral Data

Compound **49**: 1,11-oxidocalamenene: C_15_H_20_O; ^13^C-NMR (CDCl_3_, 100 MHz) data: see Table 1. EI-MS 70 eV, *m/z* (rel. int.): 216(2, M^•+^), 174(12), 173(100), 159(8), 158(62), 157(13), 156(9), 144(7), 143(39), 142(16), 141(17), 129(8), 128(26), 115(10), 43(13).

Compound **93**: atractylone: C_15_H_20_O; ^13^C-NMR (CDCl_3_, 100 MHz) data: see Table 1. EI-MS 70 eV, *m/z* (rel. int.): 217(7, M+1), 216(47, M^•+^), 201(10), 159(7), 145(13), 131(6), 121(10), 115(6), 109(14), 108(100), 105(7), 95(7), 93(13), 91(16), 79(21), 77(15), 65(6).

Compound **96**: δ-cadinen-11-ol: C_15_H_24_O; ^1^H-NMR (CDCl_3_, 400 MHz) and ^13^C-NMR (CDCl_3_, 100 MHz) data: see Table 2. HREIMS: *m/z* 220.1829 (calculated for C_15_H_24_O, 220.1827); EI-MS 70 eV, *m/z* (rel. int.): 220(1, M^•+^), 202(41, M-H_2_O), 188(10), 187(73), 174(37), 162(44), 160(17), 159(100), 147(73), 145(27), 143(12), 134(67), 133(16), 131(26), 129(16), 128(14), 120(17), 119(80), 117(20), 115(17), 107(10), 106(18), 105(69), 93(17), 92(18), 91(67), 81(16), 79(24), 77(30), 67(10), 65(13), 59(57), 55(13), 53(10), 43(20), 41(28).

Compound **101**: (*E*)-γ-bisabolen-12-al: C_15_H_22_O; ^13^C-NMR (CDCl_3_, 100 MHz) data: see Table 1. EI-MS 70 eV, *m/z* (rel. int.): 219(1, M + 1), 218(2, M^•+^), 135(41), 134(96), 121(22), 120(10), 119(54), 107(64), 105(29), 94(10), 93(100), 92(10), 91(55), 84(10), 79(49), 77(40), 67(10), 65(10), 55(33), 53(10), 43(22), 41(25).

Compound **102**: (*E*)-γ-bisabolen-12-ol: C_15_H_24_O; ^13^C-NMR (CDCl_3_, C_6_D_6_, 100 MHz) data: see Table 1. EI-MS 70 eV, *m/z* (rel. int.): EI-MS 70 eV, *m/z* (rel. int.): 221(1, M + 1), 220(2, M^•+^), 202(5, M−H_2_O), 135(33), 134(70), 132(11), 121(21), 119(33), 107(77), 94(11), 93(100), 91(46), 81(10), 79(42), 77(32), 67(10), 55(28), 53(10), 43(32).

## 4. Conclusions

Thirty oil samples were extracted and investigated using combination of chromatographic [CC, GC(RI)] and spectroscopic [GC/MS, ^13^C-NMR] techniques. In total, 103 constituents accounting for 95.7 to 99.6% of the sample compositions were identified. Fractionation by CC led to the first-time isolation from natural source and structural elucidation by 1D and 2D-NMR spectroscopy of δ-cadinen-11-ol.

*N. acuminata* leaf EO is a complex mixture characterized by a preeminence of sesquiterpenes (87.1–98.1%) exhibiting various skeletons and a tremendous chemical variability. Contents of heat-sensitive compounds such as germacrene B, furanodiene and their corresponding rearranged products, γ-elemene and curzerene were determined by combination of GC(FID) and ^13^C-NMR data. This combination of techniques ensured a correct qualitative and quantitative analysis of the thermolabile compounds.

Statistical analysis exhibited five distinct chemical groups. Groups I, II, and V were characterized by high contents of sesquiterpene hydrocarbons (respectively, M = 58.6, 55.8, and 75.3%, vs. 34.5, 30.7, and 20.6% of oxygenated sesquiterpenes) whereas groups III and IV were dominated by oxygenated sesquiterpenes-rich oils (respectively, M = 49.1 and 67.7%, vs. 42.1 and 27.0% of sesquiterpene hydrocarbons). The observed chemical variability could be related to climatic and pedologic factors.

Concerning the anti-inflammatory activity, the low ratio between the two values of IC_50_ (EO vs. NDGA) makes it possible to consider the essential oil as a high inhibitor of the LOX activity.

## Figures and Tables

**Figure 1 molecules-27-03759-f001:**
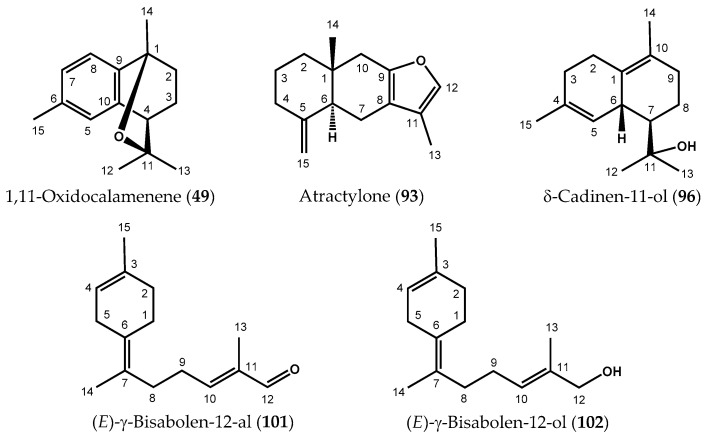
Structures of isolated compounds **49**, **93**, **96**, **101** and **102**.

**Figure 2 molecules-27-03759-f002:**
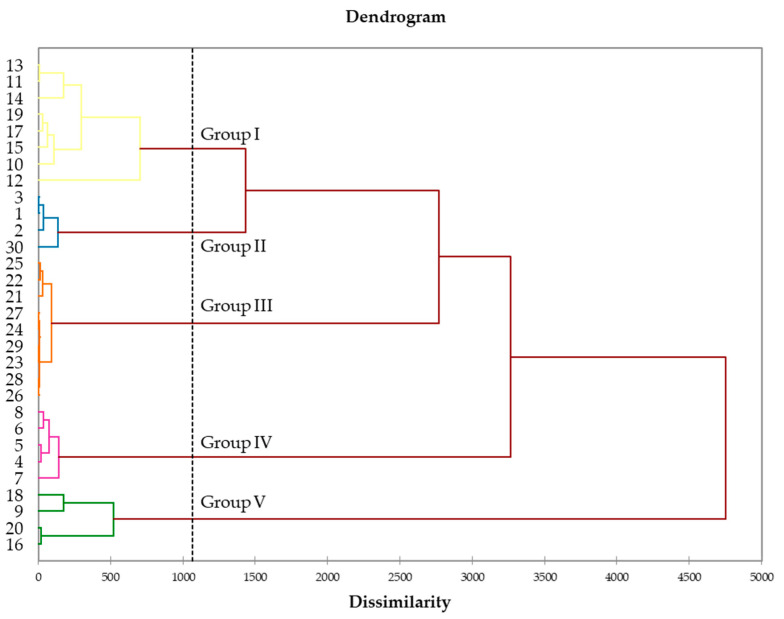
Dendrogram of hierarchical cluster analysis (HCA) of the 30 leaf EO samples from *N. acuminata*.

**Figure 3 molecules-27-03759-f003:**
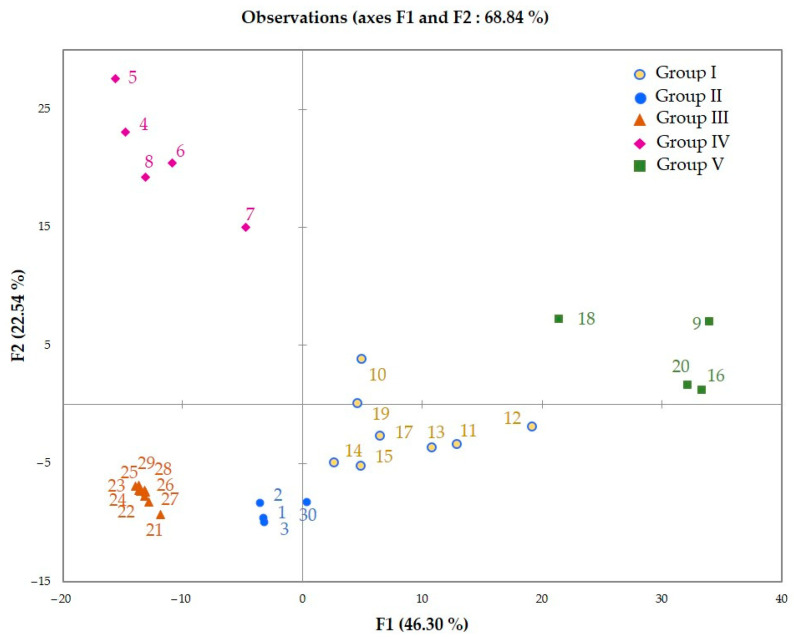
Principal component analysis (PCA) of the 30 leaf EO samples from *N. acuminata*.

**Table 1 molecules-27-03759-t001:** ^13^C-NMR data of compounds **49**, **93**, **101,** and **102**.

N °C	49		93		101		102		
	Exp	[20]	Exp	[21]	Exp	[22]	Exp	Exp *	[23] *
1	71.90	71.9	37.35	37.5	26.64	26.5	26.87	27.24	27.2
2	21.09	21.1	39.31	39.4	31.72	31.6	31.79	32.05	31.9
3	32.03	32.0	23.59	23.5	134.09	134.1	134.82	135.39	135.2
4	43.52	43.5	36.65	36.8	120.53	120.4	120.73	121.18	121.0
5	120.25	120.2	149.41	149.5	29.74	29.6	29.71	27.24	29.9
6	136.50	136.5	45.75	45.9	129.94	129.8	128.67	128.93	128.7
7	125.29	125.3	20.90	21.0	124.29	124.3	125.56	125.63	125.4
8	126.48	126.5	116.15	116.3	32.44	32.3	33.77	34.23	34.1
9	138.90	138.9	149.83	150.0	28.12	28.1	26.55	26.95	26.8
10	142.15	142.1	41.96	42.1	154.39	154.9	126.11	125.45	125.4
11	73.72	73.7	119.55	119.7	139.40	139.2	134.16	133.91	133.8
12	29.36	29.4	136.96	137.1	195.24	195.6	68.96	68.63	68.3
13	27.78	27.8	8.15	8.3	9.14	9.2	13.60	13.62	13.6
14	22.12	22.1	17.56	17.7	18.24	18.2	18.36	18.49	18.4
15	21.37	21.4	107.28	107.4	23.34	23.4	23.40	23.56	23.5

Exp: Experimental NMR data recorded in CDCl_3_; *: NMR data recorded in C_6_D_6._

**Table 2 molecules-27-03759-t002:** 1D and 2D-NMR data of δ-cadinen-11-ol (compounds **96**).

N°C	δ ^13^C (ppm)	DEPT	δ ^1^H (ppm)	Multiplicity (*J*, Hz)	COSY	HMBC	NOESY
1	130.26	C	–	–	–	–	–
2	27.45	CH_2_	α2.70	ddd (12.0, 4.5, 2.6)	2β,3	5,7,10,3,6,4,1	2β,3,9,14
			β1.98	m	2α,3	1,3,4,6,10	2α,3,6
3	32.32	CH_2_	2.04	m	3,2α	2,1,4	2α,2β,15
4	134.13	C	–	–	–	–	–
5	128.74	CH	5.71	sept (1.5)	6	15,3,6,7,1	6,15
6	38.58	CH	2.79	br s	5,7	–	2β,5,9,8β,12,13
7	49.85	CH	1.41	ddd (11.0, 8.1, 3.1)	6,8	1,5,6,8,9,11,12,13	9
8	24.53	CH_2_	α1.69	m	7,9,8β	9,6,7,11,10	8β,9
			β1.35	m	7,9,8α	6,7,10,11,12,13	6,8α,9
9	31.40	CH_2_	1.95	m	9, 8b, 8a	1,10,7,8,14	2α,6,7,8α,8β,14
10	124.12	C	–	–	–	–	–
11	74.57	C	–	–	–	–	–
12	26.58	CH_3_	1.23	s	–	8,13,7,11	6
13	29.97	CH_3_	1.29	s	–	8,12,6,7,11	6
14	18.55	CH_3_	1.67	br s	–	1,10,9	2α,9
15	23.41	CH_3_	1.64	br s	–	3,5,4	3,5

**Table 3 molecules-27-03759-t003:** Chemical composition of five leaf EO samples from *N. acuminata*.

N°	Compounds ^a^	RIl ^b^	RIa	RIp	RRF	S3 (%)	S6 (%)	S13 (%)	S20 (%)	S24 (%)	Identification
1	(*Z*)-Hex-3-en-1-ol	837	840	1388	0.826	0.1	0.1	0.1	tr	–	RI, MS
2	Hexanol	851	854	1355	0.826	0.1	–	0.1	0.1	tr	RI, MS
3	α-Thujene	932	923	1020	0.765	0.2	0.2	0.1	–	0.1	RI, MS, *^13^C-NMR*
4	α-Pinene	936	931	1016	0.765	0.1	tr	tr	tr	0.2	RI, MS, *^13^C-NMR*
5	Sabinene	973	966	1127	0.765	0.6	0.3	0.5	0.1	0.6	RI, MS, ^13^C-NMR
6	β-Pinene	978	971	1116	0.765	0.4	tr	tr	tr	0.2	RI, MS, *^13^C-NMR*
7	Myrcene	987	981	1166	0.765	0.1	0.1	–	tr	tr	RI, MS
8	α-Terpinene	1013	1010	1186	0.765	0.1	–	0.1	–	0.1	RI, MS
9	β-Phellandrene *	1023	1022	1215	0.698	–	tr	0.1	–	0.1	RI, MS
10	Limonene *	1025	1022	1205	0.765	0.1	0.1	–	tr	0.1	RI, MS
11	(*Z*)-β-Ocimene	1029	1026	1237	0.765	tr	0.1	0.1	–	0.3	RI, MS, *^13^C-NMR*
12	(*E*)-β-Ocimene	1041	1037	1255	0.765	1.2	0.8	1.5	1.7	1.2	RI, MS, ^13^C-NMR
13	γ-Terpinene	1051	1049	1250	0.765	tr	0.2	0.1	–	0.2	RI, MS, *^13^C-NMR*
14	Terpinolene	1082	1079	1288	0.765	0.2	0.1	0.1	–	0.2	RI, MS, *^13^C-NMR*
15	Linalool	1086	1085	1550	0.869	tr	0.1	tr	0.1	tr	RI, MS
16	Terpinen-4-ol	1164	1163	1604	0.869	0.1	0.1	tr	–	tr	RI, MS
17	Neral	1215	1217	1680	0.887	0.2	0.1	0.1	–	0.2	RI, MS, *^13^C-NMR*
18	Geraniol	1235	1236	1837	0.869	0.1	0.2	0.2	–	0.1	RI, MS, *^13^C-NMR*
19	Geranial	1244	1244	1732	0.887	0.1	0.1	0.1	–	0.1	RI, MS
20	Thymol	1267	1268	2190	0.808	1.3	tr	tr	tr	tr	RI, MS, ^13^C-NMR
21	Carvacrol	1278	1277	2228	0.808	0.1	0.1	tr	–	–	RI, MS
22	Cogeijerene	1285 ^c^	1282	1540	0.808	0.1	0.1	0.2	tr	tr	RI, MS, *^13^C-NMR*
23	Bicycloelemene	1338	1332	1483	0.751	0.1	tr	0.1	–	0.1	RI, MS
24	δ-Elemene	1340	1335	1472	0.751	0.4	0.2	2.2	0.2	2.6	RI, MS, ^13^C-NMR
25	α-Cubebene	1355	1348	1459	0.751	0.1	0.2	tr	–	0.1	RI, MS, *^13^C-NMR*
26	Cyclosativene	1378	1369	1483	0.751	0.1	0.1	0.1	–	0.1	RI, MS
27	α-Ylangene	1376	1371	1468	0.751	0.1	0.1	0.1	tr	–	RI, MS
28	α-Copaene	1379	1375	1493	0.751	0.1	0.1	0.1	tr	tr	RI, MS
29	β-Bourbonene	1378	1383	1520	0.751	0.1	0.1	tr	–	0.1	RI, MS
30	β-Cubebene *	1390	1387	1539	0.751	0.2	0.3	0.4	0.1	1.0	RI, MS, ^13^C-NMR
31	β-Elemene *	1389	1387	1591	0.751	4.4	0.6	1.6	4.6	1.2	RI, MS, ^13^C-NMR
32	Cyperene	1402	1399	1528	0.751	0.2	0.1	0.2	–	tr	RI, MS, *^13^C-NMR*
33	α-Gurjunene	1413	1409	1531	0.751	0.7	tr	0.2	0.1	tr	RI, MS, ^13^C-NMR
34	(*E*)-β-Caryophyllene	1421	1417	1597	0.751	1.8	3.4	20.0	34.4	1.0	RI, MS, ^13^C-NMR
35	β-Copaene	1430	1426	1591	0.751	0.5	0.1	0.7	0.1	3.1	RI, MS, ^13^C-NMR
36	γ-Elemene #	1429	1427	1640	0.751	5.5	0.3	1.3	0.3	1.1	RI, MS, ^13^C-NMR
37	*trans*-α-Bergamotene	1434	1432	1586	0.751	0.1	tr	tr	tr	0.4	RI, MS, *^13^C-NMR*
38	α-Guaiene	1440	1435	1591	0.751	tr	0.1	–	tr	0.2	RI, MS, *^13^C-NMR*
39	Sesquisabinene A	1435	1436	1647	0.751	tr	tr	tr	tr	0.6	RI, MS, ^13^C-NMR
40	Guaia-6,9-diene	1443	1437	1606	0.751	tr	tr	0.1	0.1	0.3	RI, MS, *^13^C-NMR*
41	β-Gurjunene (Calarene)	1437	1444	1591	0.751	0.1	0.1	0.1	0.1	0.2	RI, MS, *^13^C-NMR*
42	(*E*)-β-Farnesene	1446	1446	1661	0.751	0.1	0.4	0.1	tr	0.6	RI, MS, ^13^C-NMR
43	α-Humulene	1455	1450	1670	0.751	5.8	1.6	2.7	27.1	1.7	RI, MS, ^13^C-NMR
44	cis-β-Bergamotene	1435 ^d^	1452	1671	0.751	0.3	0.2	0.2	1.1	tr	RI, MS, ^13^C-NMR
45	*allo*-Aromadendrene	1462	1456	1640	0.751	0.2	tr	0.1	0.2	0.1	RI, MS, *^13^C-NMR*
46	Ishwarane	1468	1461	1645	0.751	0.1	tr	0.3	–	–	RI, MS, *^13^C-NMR*
47	γ-Muurolene	1474	1469	1688	0.751	0.1	tr	0.2	0.1	0.1	RI, MS, *^13^C-NMR*
48	4,5-*diepi*-Aristolochene	1470	1471	1705	0.751	0.3	–	tr	–	–	RI, MS, *^13^C-NMR*
49	1,11-Oxidocalamenene	1474	1472	1883	0.830	1.0	0.4	0.5	2.7	0.1	RI, MS, ^13^C-NMR
50	Germacrene D	1479	1475	1709	0.751	3.0	1.1	5.2	1.0	10.2	RI, MS, ^13^C-NMR
51	*trans*-β-Bergamotene	1480	1478	1684	0.751	0.1	tr	tr	–	0.1	RI, MS
52	β-Selinene	1486	1481	1718	0.751	0.5	0.3	0.2	0.4	0.3	RI, MS, *^13^C-NMR*
53	Furanodiene #	1485	1482	1873	0.853	0.4	0.1	0.4	0.3	3.9	RI, MS, ^13^C-NMR
54	Furano-elemene (Curzerene) #	1485	1484	1873	0.853	0.1	tr	0.1	tr	0.1	RI, MS
55	4-*epi*-Cubebol	1490	1487	1871	0.819	0.2	0.4	–	–	–	RI, MS, *^13^C-NMR*
56	Bicyclogermacrene	1494	1490	1732	0.751	0.9	0.7	0.7	0.3	0.1	RI, MS, ^13^C-NMR
57	α-Selinene	1494	1491	1723	0.751	0.7	0.2	0.4	0.2	2.2	RI, MS, ^13^C-NMR
58	α-Muurolene	1496	1494	1705	0.751	2.8	2.2	1.0	6.6	0.3	RI, MS, ^13^C-NMR
59	β-Bisabolene	1503	1500	1727	0.751	0.8	0.9	0.4	0.2	1.5	RI, MS, ^13^C-NMR
60	Cubebol	1514	1505	1885	0.819	tr	tr	0.7	tr	0.2	RI, MS, ^13^C-NMR
61	γ-Cadinene	1507	1507	1758	0.751	0.1	–	0.1	–	0.1	RI, MS
62	(*Z*)-γ-Bisabolene	1505	1510	1732	0.751	0.1	tr	tr	0.1	0.1	RI, MS
63	δ-Cadinene	1520	1514	1758	0.751	2.4	0.8	8.4	0.9	7.0	RI, MS, ^13^C-NMR
64	Kessane	1533	1521	1761	0.751	5.2	2.3	11.5	2.7	–	RI, MS, ^13^C-NMR
65	(*E*)-γ-Bisabolene	1521	1522	1758	0.751	1.1	0.4	tr	tr	3.4	RI, MS, ^13^C-NMR
66	Selina-4(15),7(11)-diene	1534	1528	1778	0.751	0.5	8.3	0.1	0.1	0.2	RI, MS, ^13^C-NMR
67	β-Elemol	1541	1534	2079	0.819	0.6	4.4	1.2	0.9	1.8	RI, MS, ^13^C-NMR
68	Selina-3,7(11)-diene	1542	1537	1778	0.751	0.2	tr	tr	tr	0.2	RI, MS, *^13^C-NMR*
69	*cis*-Cadinene ether	1551	1545	1860	0.830	–	–	0.4	–	tr	RI, MS, *^13^C-NMR*
70	(*E*)-Nerolidol	1553	1548	2042	0.819	0.5	23.3	1.6	0.7	0.2	RI, MS, ^13^C-NMR
71	Germacrene B #	1552	1551	1829	0.751	15.5	0.9	3.2	0.7	3.5	RI, MS, ^13^C-NMR
72	Palustrol	1569	1561	1924	0.819	1.5	1.2	0.2	0.1	tr	RI, MS, ^13^C-NMR
73	*cis*-Sesquisabinene hydrate	1565^e^	1564	2081	0.819	0.1	tr	0.6	0.1	tr	RI, MS, ^13^C-NMR
74	Caryophyllene oxide	1578	1570	1978	0.830	0.1	tr	0.3	2.5	tr	RI, MS, ^13^C-NMR
75	Curzerenone	1588	1575	2025	0.841	–	–	–	0.1	0.3	RI, MS, *^13^C-NMR*
76	7-*epi-cis*-Sesquisabinene hydrate	1579 ^e^	1576	2099	0.819	0.3	tr	0.7	tr	0.1	RI, MS, ^13^C-NMR
77	Viridiflorol	1592	1581	2081	0.819	0.9	tr	0.1	0.2	0.2	RI, MS, ^13^C-NMR
78	Guaiol	1593	1584	2088	0.819	0.6	12.2	2.3	1.8	11.6	RI, MS, ^13^C-NMR
79	Ledol *	1600	1593	2025	0.819	12.5	0.3	1.3	1.0	0.2	RI, MS, ^13^C-NMR
80	Copaborneol *	1595	1593	2183	0.819	1.4	tr	0.1	–	0.2	RI, MS, ^13^C-NMR
81	Eudesm-5-en-11-ol	1600 ^f^	1595	2132	0.819	tr	0.2	0.3	tr	0.8	RI, MS, ^13^C-NMR
82	*neo*-Intermedeol	1601 ^g^	1599	2146	0.819	0.1	tr	tr	tr	0.2	RI, MS, *^13^C-NMR*
83	*epi*-Cubenol	1602	1606	2048	0.819	tr	0.2	0.2	0.4	0.3	RI, MS, *^13^C-NMR*
84	Alismol	1619	1610	2248	0.830	tr	–	tr	0.5	0.1	RI, MS, *^13^C-NMR*
85	Eremoligenol	1614	1614	2196	0.819	0.1	–	–	tr	0.2	RI, MS, *^13^C-NMR*
86	10-*epi*-γ-Eudesmol	1609	1617	2096	0.819	1.4	0.3	1.3	tr	0.3	RI, MS, ^13^C-NMR
87	τ-Cadinol	1633	1625	2175	0.819	0.3	–	0.2	tr	0.4	RI, MS, *^13^C-NMR*
88	τ-Muurolol	1633	1628	2184	0.819	0.7	0.2	0.8	tr	0.7	RI, MS, ^13^C-NMR
89	α-Muurolol	1618 ^h^	1630	2212	0.819	0.2	tr	0.6	0.1	0.2	RI, MS, ^13^C-NMR
90	β-Eudesmol	1641	1634	2225	0.819	0.2	6.3	1.9	0.7	0.5	RI, MS, ^13^C-NMR
91	α-Cadinol	1643	1637	2228	0.819	1.2	0.7	0.5	tr	tr	RI, MS, ^13^C-NMR
92	α-Eudesmol	1653	1638	2216	0.819	tr	2.1	0.6	tr	tr	RI, MS, ^13^C-NMR
93	Atractylone	1652 ^i^	1639	2121	0.841	1.0	3.3	2.7	1.5	10.0	RI, MS, ^13^C-NMR
94	Intermedeol	1626 ^h^	1641	2249	0.819	0.1	0.8	0.2	0.2	tr	RI, MS, ^13^C-NMR
95	Bulnesol *	1665	1651	2207	0.819	0.4	5.2	1.5	1.0	6.2	RI, MS, ^13^C-NMR
96	δ-Cadinen-11-ol *	^j^	1651	2271	0.819	0.8	2.9	0.2	0.1	1.8	RI, MS, ^13^C-NMR
97	α-Bisabolol	1673	1666	2208	0.819	0.3	tr	0.5	0.1	0.6	RI, MS, ^13^C-NMR
98	*epi*-α-Bisabolol	1667 ^k^	1668	2214	0.819	0.2	–	0.2	–	0.1	RI, MS, ^13^C-NMR
99	Cadina-1(10),4-dien-8α-ol	1682	1671	2306	0.819	–	0.4	8.1	0.4	0.3	RI, MS, ^13^C-NMR
100	Germacrone	1684	1673	2221	0.841	0.8	0.2	tr	0.1	0.1	RI, MS, ^13^C-NMR
101	(*E*)-γ-Bisabolen-12-al	1790 ^l^	1761	2348	0.841	0.7	1.1	tr	tr	1.9	RI, MS, ^13^C-NMR
102	(*E*)-γ-Bisabolen-12-ol	^j^	1776	2549	0.819	6.1	2.5	1.0	0.3	7.1	RI, MS, ^13^C-NMR
103	(*E*)-Phytol	2114	2098	2609	0.974	0.4	0.3	0.1	0.1	0.1	RI, MS, *^13^C-NMR*
	Monoterpene hydrocarbons					3.0	1.9	2.6	1.8	3.3	
	Oxygenated monoterpenes					1.9	0.7	0.4	0.1	0.4	
	Sesquiterpene hydrocarbons					55.4	26.2	62.2	81.7	43.8	
	Oxygenated sesquiterpenes					34.8	68.7	31.3	15.8	50.7	
	Other compounds					0.6	0.4	0.3	0.2	0.1	
	**Total**					**95.7**	**97.9**	**96.8**	**99.6**	**98.3**	

^a^ Order of elution and percentages are given on an apolar column (BP-1), except components with an asterisk (*), where percentages are taken on a polar column (BP-20). (#) Thermolabile compound (Cope rearrangement under our GC conditions), percentage evaluated by a combination of GC-FID and ^13^C-NMR data [15,19]. ^b^ RIl: Retention indices reported in the Terpenoids Library Website [24] or in reference ^c^ [25]; ^d^ [25]; ^e^ [26]; ^f^ [27]; ^g^ [28]; ^h^ [29]; ^i^ [30]; ^k^ [31]; ^l^ [22]; ^j^ RI not found in literature, compounds isolated for the first time from EO. RIa, RIp: retention indices measured on apolar and polar capillary column, respectively. RRF: relative response factors calculated using methyl octanoate as internal standard (see experimental [32]). The relative proportions of constituent are expressed in g/100 g. (–): not detected; tr: traces level (<0.05%). ^13^C-NMR: compounds identified by NMR in the EO samples and obvious in at least one fraction of chromatography; *^13^C-NMR (italic)*: compounds identified by NMR in fractions of CC.

**Table 4 molecules-27-03759-t004:** Chemical variability of the main constituents of leaf essential oil from *Neuropeltis acuminata*.

Component [a]	RIa [b]	RIp [b]	Group I			Group II			Group III			Group IV			Group V		
			M% ± SD	Min	Max	M% ± SD	Min	Max	M% ± SD	Min	Max	M% ± SD	Min	Max	M% ± SD	Min	Max
β-Elemene	1387	1591	2.2 ± 0.9	1.1	3.7	4.6 ± 0.5	4.1	5.2	1.4 ± 0.2	1.1	1.7	0.8 ± 0.3	0.5	1.1	3.1 ± 1.1	2.1	4.6
(*E*)-β-Caryophyllene	1417	1597	17.6 ± 4.2	12.5	25.0	2.6 ± 1.4	1.6	4.7	1.1 ± 0.1	1.0	1.2	3.3 ± 3.6	0.9	9.4	36.6 ± 5.9	32.5	45.4
γ-Elemene #	1427	1640	1.5 ± 0.7	0.9	3.2	5.0 ± 0.8	4.1	5.8	1.2 ± 0.4	0.8	2.0	0.4 ± 0.2	0.2	0.7	0.5 ± 0.3	0.3	0.9
α-Humulene	1450	1670	2.5 ± 1.3	1.1	5.2	6.6 ± 2.1	5.0	9.7	1.3 ± 0.3	0.5	1.7	1.0 ± 0.6	0.5	1.8	18.9 ± 12.1	6.3	31.2
Germacrene D	1475	1709	4.3 ± 1.1	3.0	5.8	3.5 ± 1.4	2.4	5.6	10.6 ± 1.6	9.0	14.4	1.2 ± 0.5	0.5	2.0	1.7 ± 0.8	1.0	2.4
Furanodiene #	1482	1873	0.2 ± 0.2	tr	0.4	0.6 ± 0.7	0.1	1.6	4.0 ± 1.2	2.2	5.9	0.3 ± 0.2	0.1	0.6	0.4 ± 0.2	0.2	0.6
α-Muurolene	1494	1705	1.2 ± 0.8	0.2	2.5	2.0 ± 1.3	0.1	2.9	0.1 ± 0.1	tr	0.3	2.7 ± 0.5	2.2	3.4	6.2 ± 3.5	2.8	10.9
δ-Cadinene	1514	1758	6.2 ± 3.4	1.8	12.0	2.2 ± 0.6	1.3	2.8	7.3 ± 0.4	6.5	7.8	0.8 ± 0.3	0.5	1.2	0.9 ± 0.4	0.4	1.4
Kessane	1521	1761	10.3 ± 9.9	1.4	32.5	4.8 ± 0.6	4.2	5.5	–	–	–	1.6 ± 1.3	0.1	2.7	1.2 ± 1.1	0.4	2.7
Selina-4(15),7(11)-diene	1528	1778	1.0 ± 1.5	tr	4.3	0.5 ± 0.1	0.3	0.5	0.2 ± 0.1	tr	0.3	9.5 ± 1.7	7.4	11.2	1.1 ± 1.1	0.1	2.2
β-Elemol	1534	2079	2.2 ± 1.1	1.0	4.0	0.5 ± 0.3	tr	0.8	1.8 ± 0.3	1.5	2.5	4.5 ± 0.5	3.9	5.0	1.6 ± 0.9	0.9	2.9
(*E*)-Nerolidol	1548	2042	1.9 ± 2.1	tr	6.6	0.6 ± 0.1	0.4	0.7	0.2 ± 0.0	0.2	0.2	23.8 ± 5.4	15.8	30.8	3.0 ± 3.4	0.3	7.8
Germacrene B #	1551	1829	3.4 ± 0.8	2.2	4.8	13.4 ± 3.1	10.2	16.6	3.4 ± 1.0	2.3	5.1	0.4 ± 0.5	tr	0.9	1.3 ± 0.7	0.7	2.2
Guaiol	1584	2088	4.2 ± 2.5	2.0	7.8	0.4 ± 0.3	tr	0.7	11.4 ± 2.0	6.9	13.2	12.4 ± 1.3	10.9	13.8	2.5 ± 0.8	1.8	3.5
Ledol	1593	2025	3.4 ± 3.1	0.4	8.2	10.6 ± 2.7	7.3	13.2	0.2 ± 0.2	tr	0.6	0.6 ± 0.8	tr	1.9	1.8 ± 1.1	0.7	2.9
Atractylone	1639	2121	1.9 ± 1.2	tr	3.3	1.2 ± 1.4	tr	3.2	9.6 ± 2.3	4.8	12.1	3.0 ± 0.3	2.6	3.3	1.5 ± 0.5	0.8	2.1
Bulnesol	1651	2207	2.5 ± 1.2	1.4	4.2	0.4 ± 0.1	0.2	0.5	5.9 ± 1.2	3.2	7.2	6.5 ± 1.1	5.2	7.6	1.7 ± 0.9	1.0	3.0
Cadina-1(10),4-dien-8α-ol	1671	2306	3.6 ± 4.8	tr	12.1	–	–	–	0.3 ± 0.1	0.1	0.6	0.1 ± 0.2	tr	0.4	0.6 ± 0.7	tr	1.6
(*E*)-γ-Bisabolen-12-ol	1776	2549	1.0 ± 0.4	0.4	1.6	5.8 ± 0.4	5.2	6.1	7.3 ± 1.3	5.3	9.4	4.4 ± 2.6	2.5	8.8	0.8 ± 0.5	0.3	1.4

[a] Order of elution and percentages on apolar column (BP-1), except components with a hash (#), percentages calculated by combination of GC(FID) and ^13^C-NMR; [b] RIa, RIp: Retention indices measured on apolar and polar capillary column respectively; M% ± SD: mean percentage and standard deviation; (–): not detected; tr: traces level (<0.05%).

**Table 5 molecules-27-03759-t005:** In vitro anti-inflammatory activity of *Neuropeltis acuminata* leaf essential oil.

Anti-Inflammatory Activity (Percentage Inhibition of LOX)		IC_50_ (mg mL^−1^)	
Oil concentration (mg mL^−1^)	Inhibition (%)	Essential oil	0.059 ± 0.001
0.0125	15.20 ± 0.30	*NDGA	0.013 ± 0.003
0.0250	24.59 ± 1.42		
0.0500	47.35 ± 2.09		
0.0800	64.92 ± 2.87		
0.1000	81.87 ± 0.33		

Values are means of triplicates ± standard deviation; *NDGA: NorDihydroGuaiaretic Acid.

**Table 6 molecules-27-03759-t006:** Fractions obtained from column chromatography.

Fraction Sample	F1 P 100%	F2 P 100%	F3 P/DE 98/2	F4 P/DE 95/5	F5 P/DE 90/10	F6 P/DE 80/20	F7 DE 100%
S3 (2.605 g)	0.928	0.612	0.115	0.159	0.456	0.219	0.024
S6 (3.280 g)	0.698	0.224	0.174	0.049	1.542	0.515	0.011
S13 (2.318 g)	1.109	0.349	0.107	0.095	0.232	0.290	0.019
S20 (2.148 g)	1.311	0.423	0.155	0.019	0.131	0.037	0.014
S24 (4.110 g)	1.411	0.536	0.686	0.058	0.929	0.399	0.033

## Data Availability

The data presented in this study are available in Appendix A.

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
