# Peer review of "Neuropeltis acuminata (P. Beauv.): Investigation of the Chemical Variability and In Vitro Anti-inflammatory Activity of the Leaf Essential Oil from the Ivorian Species"

_molecules, 2022, doi:10.3390/molecules27123759_

Round 1
Reviewer 1 Report
This manuscript deals with the extraction, variability of chemical composition and determination of anti-inflammatory activity of the leaf essential oil from Neuropeltis acuminata.
This manuscript presents an interesting investigation, however before being published some grammatical errors must be corrected. In addition, there are some points that are not clear.
Abstract
-“chemical composition and variability” should be “variability of chemical composition”
“Thirty oil samples were collected from three forests….”, correct, the essential oil is "extracted" or "isolated"
-Line 21. Define initials
-Line 21. The techniques “CC, GC(RI), GC-MS and 13 C-NMR”, cannot be combined, correct
Introduction
- Define initials the first time they are named, example "CC"
Results and Discussion
Give more detail on the yields, a table could be added in supplementary material
-Explain in a better way why samples S3, S6, S13, S20 and S24 were chosen to deepen the analysis, there are other samples that have different compositions (S5, S7, S12, S18, S9), from group 3 no sample is taken.
-Line 144 in 145. samples S2, S14, S24, S44 and S47? correct
-Table S1. use 1, 2, 3, ....., for the name of the samples, define the name of the samples 1 or S1, 2 or S2,......
The discussion of most of the results is missing
Materials and Methods
To which it refers " sufficiently distant " ?
- The usefulness of the fractions presented in "Table 6. Weights of fractions from column chromatography" is not clear.
- For the analysis in NMR pure compounds are needed, it is established that the 103 compounds (there are samples with less) of each sample were divided into 7 fractions (F1-F7), groups of compounds were analyzed in the NMR?, this needs to be clarified and specified in a better way, how was the fractionation refined?
In order for the qualitative and quantitative data of chemical composition to be comparable, the analyzes of points 3.3., 3.4 and 3.5 must be carried out in the same chromatograph, or at least with the same columns (non-polar and polar).
I suggest grouping the samples in Table S1 according to their compositional relationship, to facilitate visualization.
Why was sample S24 chosen to determine the in vitro Anti-inflammatory Capacity?
Line 354. “The chemical compositions of 47 leaf essential oil sample…” 30 samples?
Conclusions
The conclusions are similar to an abstract
Author Response
Reviewer 1
This manuscript deals with the extraction, variability of chemical composition and determination of anti-inflammatory activity of the leaf essential oil from Neuropeltis acuminata.
This manuscript presents an interesting investigation, however before being published some grammatical errors must be corrected. In addition, there are some points that are not clear.
Abstract
-“chemical composition and variability” should be “variability of chemical composition”
“Thirty oil samples were collected from three forests….”, correct, the essential oil is "extracted" or "isolated"
-Line 21. Define initials
Corrections were done
Line 21. The techniques “CC, GC(RI), GC-MS and 13 C-NMR”, cannot be combined, correct
For 30 years and following a pioneering work of Formacek and Kubeczka, we developed a method based on 13C-NMR to identify terpenes in the essential oil. The identification of the individual components of essential oils was usually carried out by "on line" techniques or combination of chromatographic separation followed by spectroscopic identification of the pure compounds. We developed the direct identification of mono- and sesquiterpenes in essential oils using computer-aided analysis of 13C-NMR spectrum of the mixture, without previous separation of the components. In this method, an individual component is identified by comparison of the signals of the mixture spectrum with those of pure reference spectra present in a built library. Each compound is identified considering i) the number of observed carbons, ii) the number of overlapped signals, iii) the difference of the chemical shift of each signal in the mixture spectrum and in the reference. We applied this method in several papers about essential oils which allowed that this methodology is very well suited for the identification of stereoisomers and thermo labile compounds and accurate for intraspecific chemical variability studies.
Then, there is really a combination of techniques in this manuscript.
- For the analysis in NMR pure compounds are needed, it is established that the 103 compounds (there are samples with less) of each sample were divided into 7 fractions (F1-F7), groups of compounds were analyzed in the NMR?, this needs to be clarified and specified in a better way, how was the fractionation refined?
The major part of compounds listed in the table 3 were identified using the methodology described above.
Introduction
- Define initials the first time they are named, example "CC"
Corrections were done.
Results and Discussion
Give more detail on the yields, a table could be added in supplementary material
A table S2 was added in the supplementary material
-Explain in a better way why samples S3, S6, S13, S20 and S24 were chosen to deepen the analysis, there are other samples that have different compositions (S5, S7, S12, S18, S9), from group 3 no sample is taken
We agree that S5, S7, S12, S18, S9 are an efficient alternative but we choose the samples on the basis of three parameters;
- one sample for each group defined by statistical analysis;
- a sufficient amount of EO to realize column chromatography;
- The presence of compounds not identified by GC/MS or compounds which need an enrichment to ensure the identification.
-Line 144 in 145. samples S2, S14, S24, S44 and S47? Correct
A correction was done
-Table S1. use 1, 2, 3, ....., for the name of the samples, define the name of the samples 1 or S1, 2 or S2,......
A correction was done.
The discussion of most of the results is missing
We add a table S2 containing yields. A sentence about variation of yields in the sampling was introduced (lines 71-73). We modified several sentences in the results and discussion and conclusion parts : lines 91-92, lines 149-151, lines 185-186, line 193, lines 395-414.
Materials and Methods To which it refers " sufficiently distant " ?
A correction was done.
- The usefulness of the fractions presented in "Table 6. Weights of fractions from column chromatography" is not clear.
We modified the table 6. The last column was deleted.
In order for the qualitative and quantitative data of chemical composition to be comparable, the analyzes of points 3.3., 3.4 and 3.5 must be carried out in the same chromatograph, or at least with the same columns (non-polar and polar).
We agree. There are typographical errors in the experimental part. We correct some values (length of column, phase of column, temperature program). Obviously, e used, the same material and the same parameters for GC and GC/MS analysis.
I suggest grouping the samples in Table S1 according to their compositional relationship, to facilitate visualization.
The samples were rearranged in table S1.
Why was sample S24 chosen to determine the in vitro Anti-inflammatory Capacity?
In the previous papers concerning the anti-inflammatory activity:
- Chemical Variability and In Vitro Anti-Inflammatory Activity of Leaf Essential Oil from Ivorian Isolona dewevrei (De Wild. & T.Durand) Engl. & Diels. Molecules 2021, 26, 6228. https://doi.org/10.3390/molecules26206228
- Biological Activities and Chemical Composition of Santolina africana et Fourr. Aerial Part Essential Oil from Algeria: Occurrence of Polyacetylene Derivatives, Molecules, 2019, 24, 204; doi:10.3390/molecules24010204.
we observed that monoterpene and sesquiterpene hydrocarbon groups are probably involved in this activity. Then, we chosen a sample with a sufficient amount to prepare several concentrations (anti-inflammatory capacity) and exhibiting a high percentage of sesquiterpene hydrocarbons.
Line 354. “The chemical compositions of 47 leaf essential oil sample…” 30 samples?
The correction was done.
Conclusions
The conclusions are similar to an abstract
We rewrite the entire conclusion part.
Reviewer 2
The work of Kambirè et al. it is well structured and organized.
The experimental parts were conducted with methodological rigor and provide important and innovative results.
The work is certainly suitable for publication and only minor revisions are necessary:
-In the Abstract enter the full name of the initials CC, GC (RI), GC-MS etc as they appear for the first time in the text.
Modification made.
-Check line 276: 100g, 100g respectively. 100g is repeated twice.
Modification made.
-Table 3: correct a-guaiene with the Greek letter alpha-
Modification made.
-Table 3: it is not specified if the percentage values are the result of an average value obtained on the repetition of the analyses or if it is an absolute value. If it were an average (more correct) value, it would be advisable to enter the standard deviations.
There is no repetition. Then Mean and SD values are not defined.
Editorial Changes.
To avoid, some duplicated parts from other already published materials, we modify and rephrase several paragraphs. However, this task remains difficult. Indeed, these paragraphs reported parameters of analysis and are obviously similar to the previous papers.

Reviewer 2 Report
The work of kambirè et al. it is well structured and organized.
The experimental parts were conducted with methodological rigor and provide important and innovative results.
The work is certainly suitable for publication and only minor revisions are necessary:
-In the Abstract enter the full name of the initials CC, GC (RI), GC-MS etc as they appear for the first time in the text.
-Check line 276: 100g, 100g respectively. 100g is repeated twice.
-Table 3: correct a-guaiene with the Greek letter alpha-
-Table 3: it is not specified if the percentage values are the result of an average value obtained on the repetition of the analyses or if it is an absolute value. If it were an average (more correct) value, it would be advisable to enter the standard deviations.
Author Response

(The authors gave the same response as above.)

Round 2
Reviewer 1 Report
Errors in the manuscript have been corrected, indicated corrections have been made, and suggestions have been taken into account. In summary the manuscript has been improved, so I recommend its publication.
Author Response
No modification needed.